# Assessment of allergy knowledge among the Palestinian community: A cross-sectional study

Maha Rabayaa[1,2], Mustafa Ghanim[1*], Malik Alqub[1], Mohammad Abuawad[1], Majdi Dwikat[3], Samar Alkhaldi[3], Haneen Badawi[4], Johnny Amer[3]

1 Department of Biomedical Sciences and Basic Clinical Skills, Faculty of Medicine and Allied Medical Sciences, An-Najah National University, Nablus, Palestine, 2 Department of Physiology, Faculty of Medicine, Bolu Abant İzzet Baysal University, Bolu, Türkiye, 3 Department of Applied and Allied Medical Sciences, Faculty of Medicine and Allied Medical Sciences, An-Najah National University, Nablus, Palestine, 4 Faculty of Applied Sciences, Palestine Technical University- Kadoorie, Palestine

* Mustafa.ghanim@najah.edu

## Abstract

### Introduction

Allergy is a form of chronic illness with an increasing prevalence globally. Adequate knowledge among the community about the causes, symptoms, and treatment of allergy is crucial in preventing the associated life-threatening complications. Limited research has been conducted in Palestine regarding this health priority. The current study aimed to assess the Palestinian community's level of knowledge regarding allergy.

### Methods

An observational cross-sectional study was conducted using an online questionnaire targeting Palestinians aged 18 years and older between 1 June 2024 and 26 January 2025. The questionnaire gathered demographic information about the participants and assessed their knowledge level concerning allergy.

### Results

A total of 1002 participants were included in the study. The mean age of the participants was 30.33 years. 66.1% of the participants were females, 60.2% were unmarried, 60.1% had possessed a bachelor's degree, 63.1% had a personal history of allergy, and 82% reported knowing of someone with allergies. The mean knowledge score about allergies was 5.4 out of 10, with over half of the participants having an average level of knowledge (4–6). The knowledge score about allergies was significantly different based on the participants' sex, marital status, place of residence, educational level, and occupation (p-value <0.001).

**Data availability statement:** All relevant data are within the paper and its Supporting Information files.

**Funding:** The author(s) received no specific funding for this work.

**Competing interests:** The authors have declared that no competing interests exist.

## Conclusion

The Palestinians' knowledge of allergy is considered good regarding its common causes and symptoms. However, there is still inadequate knowledge about the treatment of allergy and its less common causes. The community awareness of allergy should be improved through targeted campaigns and brochures aimed at achieving earlier diagnosis and proper management to prevent the development of life-threatening complications.

## Introduction

Allergies are defined as a group of chronic, inflammatory disorders characterized by hypersensitivity of the immune system to certain environmental chemicals called allergens [1]. Allergens are mostly proteins or glycoproteins and are typically found in various sources, including drugs, plant pollens, animal dander, biologic products, house dust mites, foods, fungal spores, and insect venoms [2]. The etiology of allergic diseases reflects a complex and poorly understood interplay between genetic susceptibility and environmental factors [3]. Allergies vary widely in clinical manifestations and severity from mild self-limited reactions to life-threatening anaphylaxis, depending on several factors. Among these factors are the type of allergen, its size, and the route of human exposure to this allergen [2]. Indeed, allergies that are inhaled cause predominantly respiratory manifestations, while allergens that are ingested cause predominantly gastrointestinal manifestations [4]. Allergic diseases include life-threatening anaphylaxis, food allergies, certain forms of asthma, rhinitis, conjunctivitis, angioedema, urticaria, eczema, eosinophilic disorders, including eosinophilic esophagitis, and drug and insect allergies [5]. Allergies are among the most common chronic illnesses worldwide, with a prevalence that has increased [6]. Previous studies revealed that allergic diseases affect approximately 10–40% of the population [7–10]. In the United States, it was reported that nearly 1 in 3 adults and more than 1 in 4 children had experienced some form of allergy [11]. Allergic diseases constitute a considerable burden on health, social, and economic systems in both developed and developing communities [12–14]. The greatest burden of health consequences of allergies is especially seen in children, young adults, pregnant women, elders, and patients with chronic disorders [15–17].

Despite the aforementioned high global prevalence of allergies, several studies have demonstrated significant gaps in public knowledge regarding various aspects of this field [18,19]. Thse knowledge gaps put allergic patients at more risk of getting fatal anaphylactic reactions, which occur mostly in public areas. Proper management of allergy requires medical services that provide expert allergy care, and a sufficient level of public knowledge about it. Therefore, the major organizations devoted to the field of allergy strongly emphasize that educating health professionals and the public on the impact of allergic diseases as a public health concern should be encouraged [12].

In the West Bank of Palestine, where more than three million inhabitants live, little is known about the national prevalence of allergies. One study found a prevalence



of allergic rhinitis among Palestinian university students to be 3.1% [20]. However, this study was conducted on a sample from a single university. Another more recent study found that the prevalence of allergic rhinitis among Palestinian university students was 10.3% [21]. Also, this study was based on a sample from a single university, and it focused on the association of allergic rhinitis with exposure to household fuel and smoking types. These two studies found a significant gap in knowledge of allergic rhinitis among Palestinian university students and recommended the development of awareness campaigns to educate university students and the general public about allergies. To the best of our knowledge, no previous study has been conducted in Palestine on the national prevalence of allergies. Public knowledge of allergy has not been assessed in previous research. Insufficient public knowledge about allergies hinders preventive measures and is associated with delayed detection of the condition, ineffective treatment, and potentially fatal consequences for patients [22]. Therefore, this research aimed to investigate the Palestinians' knowledge about allergies.

## Methodology

### Study design

This cross-sectional study, employing a questionnaire, was undertaken between June 1, 2024, and January 26, 2025, to evaluate the Palestinian community's knowledge regarding allergies, encompassing their causes, management, and symptoms.

### Study setting

The study included a population survey of Palestinians aged 18 and older. The population included people from all the governorates in the West Bank of Palestine. Participants in the survey were chosen from a range of areas across Palestine's West Bank to ensure demographic representation. Participants were selected from local community centers, universities, and social media platforms. In order to expand the community's reach and facilitate a deeper understanding of the results of the research within the framework of Palestinian society.

### Human ethics and consent to participate

The Institutional Review Board (IRB) of An-Najah National University in Nablus, Palestine, authorized every part of the study protocol (Ref: Med. March 2024/30). The Declaration of Helsinki's rules for the use of data from humans were followed in this study. Before taking part, the participants gave their written informed consent. The informed consent form guaranteed participant anonymity and data confidentiality while outlining the study's premise.

### Study instrument

A web-based survey, conducted using Google Forms, was utilized to gather responses from participants. The questionnaire (supplementary file S1 File) contained consent forms outlining the study's aims, affirming that participation was voluntary and that data would be assessed and processed anonymously. The questionnaire was initially created and validated by Atayaa et al., utilizing the framework from the American Academy of Allergy, Asthma, and Immunology (AAAAI) and the University of Rochester Medical Center (URMC). The Arabic version of the validated questionnaire was used after getting the requested permission [19]. The questionnaire included ten items, with answers assessed based on three levels of knowledge. Each accurate response received one point, whilst erroneous responses received zero points, culminating in a maximum achievable score of 10. Participants' knowledge levels were categorized as weak (0–3), moderate [4–6], or strong [7–10] based on the overall score. The initial version of the questionnaire was created in English and thereafter translated into Arabic by proficient linguistic specialists to guarantee precision and clarity. A pilot research with 25 individuals was undertaken to evaluate internal consistency and reliability, resulting in a Cronbach's alpha coefficient of 0.722. The pilot sample size of 25 was established in compliance with recognized methodological guidelines for pilot testing [23,24]. The questionnaire comprises two sections. The initial component consists of eight questions regarding the

participant's demographic information, encompassing gender, marital status, age, level of education, residence, occupation, and prior experiences with allergies, either personally or through acquaintance. The second section encompasses inquiries regarding the etiology, manifestations, and management of allergies.

### Sampling method

The participants were recruited through many approaches, including local community centers, universities, and social media sites. Initiatives encompassed distributing the survey link extensively, utilizing researchers' networks, and interacting with the community across several platforms to enhance participation and representation, as was adopted in our previous research [25].

### Sample size

The sample size was determined using the Raosoft formula (www.raosoft.com), employing a reference proportion of 50%, a 95% confidence interval, and a 5% margin of error. The sample size was established at 385 to accurately represent the larger population and account for any non-response biases. A total of 1002 respondents engaged in the survey. Choosing an extensive sample size helps enhance analytical strength and precisely represent the variations within the study population. Although the determined sample size was less than the selected one, this decision was made to improve the quality and dependability of the results and to achieve a more comprehensive understanding of the population.

### Statistical analysis

Statistical analyses were conducted utilizing the Statistical Package for the Social Sciences Statistics (SPSS) for Windows, version 21 (IBM Corp., Armonk, N.Y., USA). Descriptive analyses, including frequency, percentages, mean, and standard deviation (SD), were conducted for the participants' fundamental characteristics, as well as the overall knowledge score, knowledge level about allergies, and correct responses for each allergy-related question. A one-way ANOVA and t-test were performed when applicable to evaluate the relationship between sample characteristics and knowledge scores about allergies. The factors deemed significant in the ANOVA test were subsequently analyzed using the post hoc Tukey test. Bivariate Pearson correlation was employed to assess the relationship between age and knowledge scores. A p-value below 0.05 was deemed statistically significant.

## Results

### Sample characteristics

A total of 1002 participants were enrolled in the study (Raw data: supplementary file S2 File). The mean age of the participants was 30.33 years (SD = 14.5). Within the total study sample, 66.1% were females, 60.2% were single, and the majority of the participants were living either in cities or villages. Regarding the participants' level of education, most of the participants had a bachelor's degree (60.1%), 8.9% had higher education, 11.3% had a diploma, 14.3% had achieved a secondary degree, and only 5.5% had less than a secondary degree. Moreover, 46.6% of the participants were students, 33.4% were employees or employers, 2.4% were retired, 6.9% were working in other jobs, and 10.7% were not working. Regarding previous experience with allergic reactions, 63.1% of the participants reported that they have had an allergy, and 82% of the participants reported that they know another person who has had an allergy. Results are shown in Table 1.

### Knowledge level about allergy among Palestinians

The mean knowledge score of the study participants about allergy was 5.4 ± 1.61. More than half of the participants achieved an average knowledge score, 30.1% achieved a strong knowledge level, and only 13.2% had weak knowledge about allergies. Results are presented in Table 2.



**Table 1. Sample characteristics (n = 1002).**

| Variable | n (%) |
|---|---|
| Age (mean ± SD) | 30.33 ± 14.5 |
| Gender | |
| Male | 340 (33.9) |
| Female | 662 (66.1) |
| Marital status | |
| Single | 603 (60.2) |
| Married | 367 (36.6) |
| Other | 32 (3.2) |
| Educational level | |
| Less than secondary school | 55 (5.5) |
| Secondary school | 143 (14.3) |
| Diploma | 113 (11.3) |
| Bachelor | 602 (60.1) |
| Higher education | 89 (8.9) |
| Have you ever had an allergy? | |
| No | 470 (46.9) |
| Yes | 532 (53.1) |
| Do you know anyone who has had an allergy? | |
| No | 180 (18) |
| Yes | 822 (82) |
| Place of residence | |
| City | 439 (43.8) |
| Village | 485 (48.4 |
| Camp | 78 (7.8) |
| Occupation | |
| Student | 467 (46.6) |
| Employee/employer | 335 (33.4) |
| Retired | 24 (2.4) |
| Other | 69 (6.9) |
| I do not work | 107 (10.7) |

**Table 2. Description of the knowledge level about allergy (n = 1002).**

| Knowledge level of allergy (score out of 10) | n (%) |
|---|---|
| Weak knowledge (0–3) | 132 (13.2) |
| Average knowledge [4–6] | 568 (56.7) |
| Strong knowledge [7–10] | 302 (30.1) |
| Total | 1002 (100) |
| Knowledge score (mean±SD) | 5.4 ± 1.61 |

Table 3 demonstrates the detailed description of the questions used to evaluate the participants' knowledge about allergies and the percentage of their correct answers for each question. More than 80% of the participants knew that anaphylaxis can occur from eating common foods such as milk, eggs, or shellfish, that the immune system is the system responsible for the allergic reaction, that the possible causes of allergy, and that allergies can cause conjunctivitis. More than 60% of the participants answered the questions about the symptoms of anaphylaxis and the duration until the



**Table 3. Evaluation of the participants' knowledge about allergies.**

| Question (True answer) | Frequency of true answer (%) |
|---|---|
| 1.  Symptoms of anaphylaxis can occur (either after a short or long period following exposure to an allergen) | 616 (61.5) |
| 2.  An anaphylactic reaction can be as simple as developing a rash after exposure to an allergen (wrong) | 65 (6.5) |
| 3.  Anaphylaxis can occur from eating common foods such as milk, eggs, or shellfish (right) | 827 (82.5) |
| 4.  Anaphylaxis always requires medical treatment (wrong) | 137 (13.7) |
| 5.  The most severe form of allergic reaction is called anaphylaxis. Which symptoms might happen with anaphylaxis? (All of the above: Difficulty in Breathing, hypotension, and rhinorrhea) | 675 (67.4) |
| 6.  If you are at risk for anaphylaxis, the best way to manage your condition is (All of the above; Avoiding allergic materials, making a plan to manage allergic cases, and always carrying an epinephrine shot) | 531 (53) |
| 7.  Which of these body systems causes allergic reactions? (the immune system) | 867 (86.5) |
| 8.  An allergen is anything that triggers an allergic response. Which of these could be an allergen? (all of the above: Dust, food, nickel) | 844 (84.2) |
| 9.  Dust mites are a common trigger for indoor respiratory allergies. Where are you most likely to find them in the home? (bed) | 40 [4] |
| 10.  Allergies can cause conjunctivitis (right) | 812 (81) |

symptoms appeared following exposure to the allergen. 53% of the participants answered the question concerning the anaphylaxis management method correctly. 13.7% of the participants knew that anaphylaxis does not always need medical treatment. 6.5% of the participants knew that the anaphylaxis reaction is not a simple rash on the skin, and only 4% of participants knew that dust mites are more likely to be found on beds.

## The association between the participants' characteristics and the allergy knowledge score

The participants' knowledge score about allergy was significantly variable based on the participants' gender, marital status, place of residence, educational level, and occupation (p-value <0.001). Females had significantly higher mean knowledge scores about allergies compared with males (5.71 ± 1.38 and 4.8 ± 1.85, respectively). Single participants had significantly higher knowledge scores compared with participants who were not single. And no significant difference in the knowledge scores between married participants and participants with other marital statuses.

According to the place of residence, participants living in cities or villages had significantly higher knowledge scores about allergies compared with those living in camps (5.56 ± 1.41, 5.43 ± 1.63, and 4.38 ± 2.16, respectively). However, no significant variation between city and village residents was observed.

Participants with high levels of education, namely bachelor's and higher education degree holders, had significantly higher knowledge scores compared with participants with lower levels of education. Additionally, Participants who have had diplomas or secondary school degrees had significantly higher knowledge scores compared with those who have not even completed the secondary school level. There were no significant differences in the knowledge scores between bachelor's degree and higher education degree holders and between diplomas and secondary school degree holders. The knowledge scores about allergies were significantly variable based on the occupation of the participants. Participants who were students or employees/employers had significantly higher knowledge about allergies compared with the participants who had other jobs and those who were not working.

The knowledge scores about allergies are also affected by the participants' experience with allergies. Significantly higher knowledge scores were observed among participants who had an allergic reaction than those with no allergies (5.58 ± 1.49 and 5.2 ± 1.72, p-value <0.001). Additionally, participants who know other people who have had allergies showed significantly higher knowledge scores about allergies compared with those who do not know anyone with allergies (5.55 ± 1.49 and 4.72 ± 1.94, p-value <0.001).

 

In addition, a significant negative correlation was observed between the participants' knowledge score about allergy and age (Pearson correlation −0.367, p-value <0.001). Results are shown in Table 4.

## Discussion

Allergic reactions are a growing public health concern with variable degrees of symptoms and occur more frequently in children than adults [26]. The severity of allergic reaction could be compounded by late diagnosis and improper treatment. Therefore, enhacing community awareness about allergies could decrease the incidence of severe cases, as early detection can prevent the development of life-threatening symptoms [27].

In the current study, more than half of the participants reported a history of allergic reactions, which is consistent with global increases in allergy conditions. Additionally, 82% of the participants reported having contact with individuals suffering from allergies, reflecting an increase in prevalence of allergic reactions within the Palestinian community. A previous study conducted on school children in a Palestinian city reported a prevalence of allergy of 12.3% based on the measurement of the total and allergen-specific IgE concentrations, with the highest reported prevalence of allergies against dogs, dust mites, and milk [28]. Another study in Palestine reported that the prevalence of allergic rhinitis was 10.3% among university students [21]. In the Middle East, the prevalence of self-reported or parent-reported symptoms of allergic rhinitis ranged from 9% to 38% [29]. In the United States, 99% the primary care physicians reported providing care for food-allergic patients. These findings illustrate that allergies, regardless of their causes, are a common clinical observation that should be a focus area for awareness, in order to facilitate their rapid diagnosis and treatment.

The current findings revealed that the Palestinian community is well-informed about allergy, with most of them having a mean knowledge score of 5.4 out of 10. The Palestinian community's knowledge score about allergy is higher than that reported among Syrian hospital patients using the same study tool, where the mean test score was 3.94 out of 10 [19]. These variations could be explained partly by the targeted group of the population, since our study targeted the Palestinians in general, while the Syrian study included only hospital patients, who might be mostly old people (the mean age: 43.2 years), and thus less knowledgeable about allergy. In addition, weak public knowledge about allergies has been reported in Syria [19], Germany [30,31], Turkey [32], and Saudi Arabia [18].

The majority of participants (82.5%) accurately identified milk, eggs, and shellfish as causes for anaphylaxis. This is consistent with global epidemiological data, since these foods are among the most prevalent causes of severe allergic reactions, particularly in children and people with IgE-mediated hypersensitivity [33]. Public health initiatives and regulatory labeling rules (e.g., in the United States) have probably contributed to this high awareness by publicizing certain allergens [34,35]. The current study participants' strong performance on basic allergy facts (symptoms, triggers, and immune basis) is consistent with existing literature showing sufficient awareness in these domains [18]. Only 53% were aware of the optimal steps (avoiding, planning, and carrying epinephrine injection). This gap of knowledge is concerning because delayed or inappropriate care increases morbidity and mortality risks. Furthermore, other research has revealed similar findings; very few knew that not all anaphylactic reactions necessarily require immediate medical treatment (13.7%) or that anaphylaxis can involve more than just a rash (6.5%). Takrouni et al. also reported that the public's grasp of allergy management is weak [18]. These misconceptions reflect a broader public confusion about allergy severity and self-management, as noted in surveys of health literacy [34]. A study on healthcare professionals revealed that 70% of the participants knew the exact dose of epinephrine in the treatment of anaphylaxis, with a significant lower level of knowledge among allied healthcare professionals [32]. These findings are alarming, as epinephrine injections remain the only option available to treat food-induced anaphylaxis. According to previous studies, the use of epinephrine injections has resulted in the stabilization of mortality rates associated with food-induced anaphylaxis [36–39]. Sampson et al. noticed that fatal anaphylactic reactions were associated with delayed use of epinephrine injections for at least 30 minutes following the onset of anaphylaxis [40]. Thus, education campaigns have to be established to increase the awareness of these injections in the management of food-induced anaphylaxis. Only 4% knew dust mites are commonly found in beds, a



**Table 4. The association between the participants' characteristics and allergy knowledge score (n = 1002).**

| Variable | mean | SD | p-value | Pairwise comparison | 95% Confidence interval | |
|---|---|---|---|---|---|---|
| | | | | | Lower | Upper |
| Gender | | | | | | |
| Male | 4.8 | 1.85 | <0.001 | Male vs female | −1.12 | −0.71 |
| Female | 5.71 | 1.38 | | | | |
| Marital status | | | | | | |
| Single | 5.7[a] | 1.36 | <0.001 | Single vs married | 0.48 | 0.97 |
| Married | 4.98[b] | 1.83 | | Single vs others | 0.44 | 1.78 |
| Others | 4.59[b] | 1.93 | | Married vs others | −0.29 | 1.06 |
| Educational level | | | | | | |
| Less than secondary school | 3.96[a] | 2.04 | <0.001 | Less than secondary vs secondary school | −1.77 | −0.44 |
| Secondary school | 5.07[b] | 1.85 | | Less than secondary vs diploma | −1.48 | −0.10 |
| Diploma | 4.75[b] | 1.79 | | Less than secondary vs bachelor | −2.31 | −1.13 |
| Bachelor | 5.68[c] | 1.38 | | Less than secondary vs higher education | −2.51 | −1.07 |
| Higher education | 5.75[c] | 1.34 | | Secondary school vs diploma | −0.21 | 0.85 |
| | | | | Secondary school vs bachelor | −1.01 | −0.22 |
| | | | | Secondary school vs higher education | −1.25 | −0.11 |
| | | | | Diploma vs bachelor | −1.36 | −0.50 |
| | | | | Diploma vs higher education | −1.60 | −0.40 |
| | | | | Bachelor vs higher education | −0.55 | 0.41 |
| Have you ever had an allergy? | | | | | | |
| No | 5.2 | 1.72 | <0.001 | No vs yes | −0.57 | −0.18 |
| Yes | 5.58 | 1.49 | | | | |
| Do you know anyone who has had an allergy? | | | | | | |
| No | 4.72 | 1.94 | <0.001 | No vs yes | −1.09 | −0.58 |
| Yes | 5.55 | 1.49 | | | | |
| Place of residence | | | | | | |
| City | 5.56[a] | 1.41 | <0.001 | City vs village | −0.12 | 0.37 |
| Village | 5.43[a] | 1.63 | | City vs camp | 0.71 | 1.63 |
| Camp | 4.38[b] | 2.16 | | Village vs camp | 0.59 | 1.50 |
| Occupation | | | | | | |
| Student | 5.75[a] | 1.31 | <0.001 | Student vs employee/employer | −0.08 | 0.52 |
| Employee/employer | 5.53[a] | 1.52 | | Student vs retired | −0.17 | 1.58 |
| Retired | 5.04[ac] | 2.1 | | Student vs other | 1.25 | 2.33 |
| Other | 3.96[b] | 2 | | Student vs I do not work | 0.77 | 1.66 |
| I do not work | 4.53[bc] | 1.85 | | Employee/employer vs retired | −0.40 | 1.36 |
| | | | | Employee/employer vs other | 1.02 | 2.12 |
| | | | | Employee/employer vs I do not work | 0.53 | 1.45 |
| | | | | Retired vs other | 0.10 | 2.07 |
| | | | | Retired vs I do not work | −0.43 | 1.45 |
| | | | | Other vs I do not work | −1.22 | 0.07 |
| Age (Pearson correlation) | −0.367** | | <0.001 | | | |

**Correlation is significant at the 0.01 level (2-tailed).

critical oversight given their role in perennial allergic asthma and rhinitis [41,42]. This demonstrates the shortage of public education about indoor allergy avoidance techniques. The extremely low awareness of dust mites as an allergen source indicates that the majority of participants were unfamiliar with this common allergen. This is consistent with the data that public knowledge frequently concentrates on food or seasonal allergens, with less emphasis on persistent indoor dangers. Such ignorance of significant allergen sources may contribute to uncontrolled exposures (e.g., failure to apply dust-control measures) and an increased allergic burden. The current results revealed variation in allergy-related knowledge. These gaps highlight critical priorities for targeted education and underscore the need for standardized public awareness campaigns to improve allergy literacy.

Our results demonstrated significant associations between demographic variables and the knowledge score of allergies. For instance, females have a higher knowledge score compared to males. This could be attributed to that females are more interested in nutritional and preventive health behaviors than males. A study conducted in Kuwait among kindergarten teachers revealed that female teachers are more likely to recognize allergy symptoms and are more adherent to safety protocols [43]. Despite fewer studies comparing gender groups with allergy knowledge scores, the gender gaps could also be attributed to a societal role where females are more responsible for childcare and their health, especially in Arabic culture.

Regarding the educational levels, higher education is associated with better knowledge scores about allergy, which is consistent with global trends. For example, a study conducted in Arab countries like Kuwait reported that trained teachers have better knowledge scores than untrained teachers [43]. In addition, a study conducted in Syria revealed that university-level students have a better food knowledge score than those with elementary education [19]. In the USA, a study reported that teachers with previous training in food allergy have higher knowledge scores than those without training [44]. Moreover, a cross-sectional study conducted among restaurant staff in Germany, where knowledge scores were suboptimal, reported that the level of education influences the knowledge scores in food allergy [31]. Thus, our finding suggests that proper and higher levels of education and training are necessary for better knowledge of food allergy. Participants with an allergy or who knew someone with allergies had a higher knowledge score. This result is aligned with a study conducted among parents with allergic children who developed detailed knowledge about food allergy [45]. On the other hand, a previous study revealed that Korean children with and without food allergy experience had a similar level of knowledge on food allergies, and the children with food allergy experience thought that food allergy-related labeling on school menus was not informative [30]. The current study revealed a good knowledge of allergy among the Palestinian community, especially concerning the common causes and the associated symptoms. The current study is considered comprehensive in its topic because most previous studies were concerned with just a specific form of allergy, such as drug allergy, food allergy, or allergic rhinitis. The study included a large number of participants with a strong response rate and high-quality and integrity of their answers. These can be partially attributed to the anonymity of the online questionnaire and the ethical safeguards, particularly reporting the IRB approval and the informed consent.

## Limitations

The current study has important limitations that should be considered. The cross-sectional nature of the study limits its ability to establish causality. It is worth mentioning that the study sample was skewed toward younger and well-educated participants. The use of online distribution of the questionnaire might exclude vulnerable groups, most at risk of poor allergy awareness, because of limited internet access or their inability to participate [46] and individuals possessing a heightened interest in or comprehension of food allergies may have been more inclined to engage [47]. All these factors might partly explain the good knowledge of allergies in the current study, and limit the generalizability of the conclusions because illiterate, elderly and rural populations might not be well represented. It was reported in previous studies that older age, illiterate and rural populations were were



associated with lower knowledge of allergies [18,19,48]. Thus, conclusions of the current study should be trated with caution.

## Conclusions

The current study revealed that the Palestinian community demonstrates a satisfactory, though not comprehensive, level of knowledge regarding allergies. Observations have shown that the participants have good knowledge about the common causes and symptoms of allergy. However, there were some gaps in their knowledge, such as recognizing dust mites as a cause of allergy and the proper management of allergy cases. Awareness campaigns and educational programs must be developed to educate the general public about allergens and allergic diseases. Educational interventions for different parts of the community using appropriate methods targeting children, teachers, and families could meaningfully improve the community's knowledge and awareness about allergy. Appropriate and timely interventions must be included in the awareness programs; thus, non-health professionals could seek the required interventions and avoid developing life-threatening conditions.

## Supporting information

**S1 File. The questionnaire of the study.**
(PDF)

**S2 Table. The raw data of the study.**
(XLSX)

## Acknowledgments

The authors acknowledge the Faculty of Medicine and Health Sciences at An-Najah National University in Palestine (www.najah.edu) for the technical support provided to publish the present manuscript. We would like to express our gratitude to Dr. Waleed Salameh, an expert in Educational English from the Faculty of Graduate Studies at An-Najah National University, for his assistance with the English editing of the revised manuscript.

## Author contributions

**Conceptualization:** Mustafa Ghanim, Maha Rabayaa.

**Data curation:** Mustafa Ghanim, Maha Rabayaa, Malik Alqub, Mohammad Abuawad, Haneen Badawi, Johnny Amer.

**Formal analysis:** Mustafa Ghanim, Maha Rabayaa, Malik Alqub, Majdi Dwikat, Samar Alkhaldi, Haneen Badawi.

**Investigation:** Mustafa Ghanim, Maha Rabayaa, Mohammad Abuawad, Haneen Badawi, Johnny Amer.

**Methodology:** Mustafa Ghanim, Maha Rabayaa, Malik Alqub, Mohammad Abuawad, Majdi Dwikat, Samar Alkhaldi, Haneen Badawi.

**Project administration:** Mustafa Ghanim, Maha Rabayaa, Johnny Amer.

**Supervision:** Mustafa Ghanim, Samar Alkhaldi.

**Validation:** Mustafa Ghanim, Maha Rabayaa, Malik Alqub, Mohammad Abuawad, Majdi Dwikat.

**Visualization:** Mustafa Ghanim.

**Writing – original draft:** Mustafa Ghanim, Maha Rabayaa, Malik Alqub, Mohammad Abuawad, Majdi Dwikat, Samar Alkhaldi, Haneen Badawi, Johnny Amer.

**Writing – review & editing:** Mustafa Ghanim, Maha Rabayaa, Malik Alqub, Mohammad Abuawad, Majdi Dwikat, Samar Alkhaldi, Haneen Badawi, Johnny Amer.



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
