## [Decision Letter · Decision Letter 0]

21 Oct 2025

Dear Dr. Ghanim,

Thank you for submitting your manuscript to PLOS ONE. After careful consideration, we feel that it has merit but does not fully meet PLOS ONE’s publication criteria as it currently stands. Therefore, we invite you to submit a revised version of the manuscript that addresses the points raised during the review process.

We look forward to receiving your revised manuscript.

Kind regards,

Sanket Kaushik, PhD

Academic Editor

PLOS ONE

Journal Requirements:

https://pubmed.ncbi.nlm.nih.gov/39910132/

In your revision ensure you cite all your sources (including your own works), and quote or rephrase any duplicated text outside the methods section. Further consideration is dependent on these concerns being addressed.

4. We note that there is identifying data in the Supporting Information file <Allergy Raw data.xlsx>. Due to the inclusion of these potentially identifying data, we have removed this file from your file inventory. Prior to sharing human research participant data, authors should consult with an ethics committee to ensure data are shared in accordance with participant consent and all applicable local laws.

-Location data

Reviewers' comments:

Reviewer's Responses to Questions

**Comments to the Author**

1. Is the manuscript technically sound, and do the data support the conclusions?

Reviewer #1: Yes

Reviewer #2: Yes

2. Has the statistical analysis been performed appropriately and rigorously?

Reviewer #1: Yes

Reviewer #2: Yes

3. Have the authors made all data underlying the findings in their manuscript fully available?

Reviewer #1: No

Reviewer #2: Yes

4. Is the manuscript presented in an intelligible fashion and written in standard English?

Reviewer #1: Yes

Reviewer #2: Yes

Reviewer #1: 1. In the abstract, start the second sentence with " adequate instead of proper

2. Spell check limited

3. Average, good, inadequate, insufficient, and weak were the terminologies used to describe or define the level of knowledge. Better to use either adequate and inadequate or sufficient and insufficient or good and poor or strong and weak to avoid confusion

4. Introduction well written

5. The study aims to estimate the prevalence, in addition to knowledge assessment, but not shown in the results

6. Justify the reasons for selecting 25 people for pilot study

7. Table 2 to be revised

Reviewer #2: I congratulate you for this important work. This study assessed allergy-related knowledge among 1,002 Palestinians aged 18 years and older using a validated online questionnaire. The mean knowledge score was 5.4/10, with most participants demonstrating average awareness of allergy causes and symptoms but notable gaps in understanding management strategies, such as the use of epinephrine and recognition of dust mites as a major allergen. Knowledge levels varied significantly by gender, education, occupation, and prior personal or social experience with allergy. The authors conclude that while general awareness is fair, targeted educational campaigns are needed to address critical misconceptions and improve community preparedness for allergy-related emergencies. some points for improvment:

The sample is skewed toward younger and well-educated participants. Please expand discussion on how this limits generalizability, particularly for older, rural, or less literate populations. Highlight how online distribution might exclude vulnerable groups most at risk of poor allergy awareness. The finding that only 4% recognized dust mites as a common allergen and that just over half understood anaphylaxis management is critical. These points should be emphasized further as priorities for targeted public health education. Consider linking these gaps to regional healthcare challenges, e.g., access to epinephrine autoinjectors. While p-values are reported, consider also including effect sizes or confidence intervals to quantify differences more meaningfully. The IRB approval and informed consent are clearly reported. It might be helpful to briefly state in the discussion how ethical safeguards strengthened participant trust and response quality.

**Do you want your identity to be public for this peer review?** For information about this choice, including consent withdrawal, please see our Privacy Policy

Reviewer #1: No

Reviewer #2: **Yes: ** Dr Daniel Elbirt

---

## [Author Response · Author response to Decision Letter 1]

2 Nov 2025

Dear respected editor,

We would like to thank you and thank the reviewers for your valuable comments and suggestions which enhanced the quality of our manuscript. Kindly find below our responses and we highlighted these corrections in red in the revised manuscript.

Kind regards,

Muastafa Ghanim, Corresponding author

Journal Requirements:

Done.

https://pubmed.ncbi.nlm.nih.gov/39910132/

In your revision ensure you cite all your sources (including your own works), and quote or rephrase any duplicated text outside the methods section. Further consideration is dependent on these concerns being addressed.

Any duplicated sentences were rephrased.

Done.

3. We note that there is identifying data in the Supporting Information file <Allergy Raw data.xlsx>. Due to the inclusion of these potentially identifying data, we have removed this file from your file inventory. Prior to sharing human research participant data, authors should consult with an ethics committee to ensure data are shared in accordance with participant consent and all applicable local laws.

-Location data

Done.

Done.

The reference list was reviewed and any errors were corrected.

Reviewer #1:

1. In the abstract, start the second sentence with " adequate instead of proper

Response: Correction is done.

2. Spell check limited

Response: Thank you for your comment. The text of all of the manuscript was reviewed by authors for any typo and linguistic errors. Also, the manuscript was edited by our colleague Dr. Waleed Salameh, an expert in Educational English from the Faculty of Graduate Studies at An-Najah National University, for whom we added acknowledgement at the end of the manuscript.

3. Average, good, inadequate, insufficient, and weak were the terminologies used to describe or define the level of knowledge. Better to use either adequate and inadequate or sufficient and insufficient or good and poor or strong and weak to avoid confusion

Response: Thank you for your comment. The correction was done. The words: strong, average, and weak are used throughout the manuscript to describe the knowledge levels to make it uniform and prevent confusion. The knowledge levels are divided in the manuscript into three levels based on the study tool description.

4. Introduction well written

Thanks for your support.

5. The study aims to estimate the prevalence, in addition to knowledge assessment, but not shown in the results

Response: Thank you for your notice. The study aimed to evaluate the knowledge about allergies among Palestinians. We have noticed that prevalence is used once at the end of the introduction, and proper correction is done.

6. Justify the reasons for selecting 25 people for pilot study.

Response: The sample size of 25 participants was selected based on established recommendations for pilot studies, where sample sizes of anywhere from 12 to 35 have been proposed, and it is considered sufficient for estimating parameters in questionnaire-based studies. Proper justification with appropriate references has been added to the methods section in the manuscript.

7. Table 2 to be revised.

Response: Table 2 is revised.

Reviewer #2:

I congratulate you for this important work. This study assessed allergy-related knowledge among 1,002 Palestinians aged 18 years and older using a validated online questionnaire. The mean knowledge score was 5.4/10, with most participants demonstrating average awareness of allergy causes and symptoms but notable gaps in understanding management strategies, such as the use of epinephrine and recognition of dust mites as a major allergen. Knowledge levels varied significantly by gender, education, occupation, and prior personal or social experience with allergy. The authors conclude that while general awareness is fair, targeted educational campaigns are needed to address critical misconceptions and improve community preparedness for allergy-related emergencies. some points for improvment:

1. The sample is skewed toward younger and well-educated participants. Please expand discussion on how this limits generalizability, particularly for older, rural, or less literate populations. Highlight how online distribution might exclude vulnerable groups most at risk of poor allergy awareness.

Response: Correction is done, the discussion has been expanded to acknowledge the limitations in sample representativeness.

2. The finding that only 4% recognized dust mites as a common allergen and that just over half understood anaphylaxis management is critical. These points should be emphasized further as priorities for targeted public health education. Consider linking these gaps to regional healthcare challenges, e.g., access to epinephrine autoinjectors.

Response: Correction is done.

3. While p-values are reported, consider also including effect sizes or confidence intervals to quantify differences more meaningfully.

Response: Thank you for your suggestion. The 95% confidence intervals of the difference are added to Table 4.

4. The IRB approval and informed consent are clearly reported. It might be helpful to briefly state in the discussion how ethical safeguards strengthened participant trust and response quality.

Done.

---

## [Editor Report · Decision Letter 1]

2 Dec 2025

Assessment of Allergy Knowledge Among the Palestinian Community: A Cross-Sectional Study

PONE-D-25-38756R1

Dear Dr. Ghanim,

We’re pleased to inform you that your manuscript has been judged scientifically suitable for publication and will be formally accepted for publication once it meets all outstanding technical requirements.

Kind regards,

Sanket Kaushik, PhD

Academic Editor

PLOS ONE
---

## [Editor Report · Acceptance letter]

PONE-D-25-38756R1

PLOS One

Dear Dr. Ghanim,

I'm pleased to inform you that your manuscript has been deemed suitable for publication in PLOS One. Congratulations! Your manuscript is now being handed over to our production team.

Kind regards,

on behalf of

Dr. Sanket Kaushik

Academic Editor

PLOS One